# Extraforaminal Full-Endoscopic Approach for the Treatment of Lateral Compressive Diseases of the Lumbar Spine

**DOI:** 10.3390/jpm13030453

**Published:** 2023-02-28

**Authors:** João Paulo Machado Bergamaschi, Kelsen de Oliveira Teixeira, Thiago Queiroz Soares, Fernando Flores de Araújo, Gustavo Vitelli Depieri, Ariel Falbel Lugão, Rangel Roberto de Assis, Ricardo Squiapati Graciano, Luiz Henrique Dias Sandon, Esthael Cristina Querido Avelar Bergamaschi, Herton Rodrigo Tavares Costa, Helton Luiz Aparecido Defino

**Affiliations:** 1Atualli Spine Care Clinic, Sao Paulo 01419-001, SP, Brazil; 2Samaritan Hospital in Sao Paulo, Sao Paulo 01232-010, SP, Brazil; 3Hospital das Clínicas of Ribeirao Preto Medical School, Sao Paulo University, Ribeirao Preto 14015-010, SP, Brazil

**Keywords:** spinal stenosis, spine, injuries, hernia, percutaneous diskectomies

## Abstract

Background: The authors conducted a 2-year retrospective follow-up to investigate the efficiency of an extraforaminal full-endoscopic approach with foraminoplasty used to treat lateral compressive diseases of the lumbar spine in 247 patients. Methods: The visual analogue scale (VAS), Oswestry disability index (ODI), and MacNab scale were used to analyze the results collected during the preoperative and postoperative periods. Results: The most common diagnosis was disk herniation with lateral recess stenosis, and the most common surgical level among patients was between L4 and L5 on the left side. Pain decreased over time, as determined during sessions held to evaluate pain in the lumbar, gluteal, led, and foot regions. The ODI demonstrated significant enhancement over the evaluation period and the MacNab scale classified the surgery as good or excellent. The most common complication was dysesthesia. Conclusions: An extraforaminal full-endoscopic approach with foraminoplasty can be recommended in cases of lateral herniation or stenosis for patients with symptoms of radiculopathy, and for those who have not responded to conventional rehabilitation treatment or chronic pain management. Few complications arose as a result of this approach, and most of them were treated clinically.

## 1. Introduction

The lumbar transforaminal endoscopic approach originates from Erlacher’s posterolateral approach to nucleography [1]. In 1975, Hijikata described the first percutaneous nucleotomy achieved via this approach using a 2.3 mm cannula and 2.1 mm forceps [2]. In the 1980s, Kambin and Gellman [3] and Kambin and Sampson [4] improved this technique. They described the Kambin triangle, formed by the emerging root, the proximal plateau of the caudal vertebra, and the descending root/dural sac, as a safe area for a transforaminal approach to the disk. Based on the evolution of this technique, Onik et al. [5] developed an automated percutaneous diskectomy.

Mayer et al. [6] pioneered the use of an imaging system to perform percutaneous diskectomy, despite the limited image quality. In addition, the principle of continuous irrigation and suction has not yet been applied to the spine because of the erroneous assumption that this process may not be useful or necessary in non-preformed anatomical space cases of joints. In the 1980s and 1990s, the spectrum of indications for posterolateral and transforaminal endoscopic techniques was still limited, which is one reason why endoscopic diskectomy has remained at a low level of acceptance among spine surgeons. In addition, other criteria, such as limited optical instruments and systems, were used as a justification for this fact, in addition to the unproven technical advantage of this method compared to microsurgery.

In the late 1990s, based on the description of the Yeung Endoscopic Spine System (YESS), the multichannel endoscope with continuous irrigation of saline reduced the rates of intra- and postoperative bleeding and infections, significantly improving the visibility of anatomical structures [7]. Yeung’s system advocated the inside-out technique for a transforaminal approach. At the same time, Hooglan and Scheckenbach described the inside-out technique for transforaminal diskectomy based on the development of the Thomas Endoscopic Spine System (TESSYS) [8]. Schubert and Hoogland [9] described a percutaneous foraminoplasty system with progressive cutters using fluoroscopy to expand the range of indications for transforaminal endoscopy. These cutters yielded the opening of the intervertebral foramen and the subsequent use of the endoscopy system to perform the transforaminal diskectomy. In 2007, Ruetten et al. described new possibilities for an endoscopic approach to spinal treatment, such as an extreme lateral approach to central hernias, improving the endoscope system with larger working channels, and allowing more accurate and efficient instruments for diskectomy and osteoligamentous decompression [10]. Since then, there has been significant interest in the development of transforaminal techniques with greater efficiency and reproducibility among spine surgeons, in addition to improvement of patient safety. Thus, new possibilities for approaching the intervertebral foramen have emerged, such as the extraforaminal approach. Specifically, for extraforaminal hernias, Choi et al. [11] described the extraforaminal disk approach with an entry point in the skin more medial than the classic posterolateral transforaminal approach. This technique’s objective was to directly approach the extraforaminal hernia, without the need to approach the foramen or the centrolateral portion of the disk [11]. However, the largest extraforaminal hernias usually move from the emerging root to the side. In addition, this technique may provide a greater risk of compression or root damage in minor extraforaminal hernias because of the possibility that the emerging root is located on the hernia.

The lower risk of dysesthesia supports the idea of not directly accessing the foramen due to the manipulation of the emerging root in the postoperative period. Based on a study of 33 cases, Ahn et al. suggested that an extraforaminal approach might result in less dysesthesia in the postoperative period [12]. However, this extraforaminal approach can cause unintentional opening or damage to the facet joint. Because of this, Ruetten and Komp [13] indicated the use of a lumbar pedicular extraforaminal approach previously described for the thoracic spine [14]. For this reason, this strategy has emerged as an option to reduce traumatic damage to emerging roots. However, studies evaluating this strategy are scarce. Additionally, the Kambin safety triangle’s significant anatomical variation can generate manipulation or damage to the emerging root, even when taking an extraforaminal pedicular approach [15]. In this context, this study has the purpose of investigating the efficacy of the extraforaminal full-endoscopic approach (Figure 1) with foraminoplasty in the treatment of lateral compressive diseases of the lumbar spine.

## 2. Materials and Methods

The inclusion criteria for patients participating this retrospective study were: (1) unilateral root pain compatible with the site of compression on magnetic resonance imaging; (2) pain of intensity equal to or greater than five on the visual analog pain scale (VAS); (3) moderate or severe functional disability according to the Oswestry Disability Index (ODI); and (4) failure of conservative treatment after 6 weeks. Patients with at least one of the following criteria were excluded from the study: (1) segmental instability (sliding > 3 mm or inclination more significant than 15° of the vertebral plateaus on dynamic radiographs); (2) severe lumbar stenosis (spinal canal < 10 mm in diameter in the sagittal section and <70 mm^2^ in the area in the axial section); (3) previous surgery on the lumbar spine; (4) cauda equina syndrome; (5) tumor; (6) trauma; or (7) infection.

The Samaritano Hospital Ethics Committee approved this study under protocol 35420420.5.0000.5487. All the patients signed an informed consent form. The full-endoscopic extraforaminal surgical technique treated 247 patients with foraminoplasty between August 2016 and August 2018.

### 2.1. Surgical Procedure

The patients were kept under conscious sedation in a prone position on a radiolucent table for the surgical procedure, with their hips and knees partially flexed. The marking of the entry point on the skin was confirmed with the image intensifier’s aid, laterally, approximately 11–12 cm from the midline. Local anesthesia was performed with 1% lidocaine without a vasoconstrictor in the following structures: skin; subcutaneous tissue; paravertebral musculature; and peri-facet region. Initially, we introduced the work sleeve, and under direct supervision, we performed the foraminoplasty with a drill. We believe that this makes the procedure safer and allows it to be performed in cases in which the Kambin’s triangle is very small. A 18-gauge needle was then introduced to the lateral portion of the superior articular process (SAP) of the caudal vertebra. The final target point was confirmed using fluoroscopy in the anteroposterior and lateral views. From there, a guidewire, dilator, and beveled working cannula were inserted at 70°; subsequently, the 30° inclination endoscope and 4.2 mm working channel, accompanied by continuous saline irrigation, were introduced [17]. The opening of the working cannula was supported by the SAP of the caudal vertebra (Figure 2). Bipolar electrocautery was used to perform hemostasis; it was connected to the Surgimax device. A foraminoplasty cutting drill was used to remove the bone from part of the SAP of the caudal vertebra. The foraminal ligaments and lateral portion of the yellow ligament were removed using a rotating Kerrison or endoscopic forceps. After foraminoplasty, the cannula advanced towards the disk, which had its herniated fragment easily identified and removed with the aid of endoscopic disk forceps (outside-in technique). In cases of bone stenosis only, the disk was not addressed (outside-out approach). Foraminoplasty and diskectomy were performed after direct visualization of the decompressed root in the foramen (emergent root) or lateral recess (transversing root). After the procedure, all patients were allocated to a continuous physical rehabilitation program and monitored for 24 months.

### 2.2. Data Analyses

Patients were evaluated using the following instruments: VAS, ODI, and MacNab scale, in the preoperative moments (T0) and 2 days (T1), 1 month (T2), 3 months (T3), 6 months (T4), 12 months (T5), and 24 months (T6) postoperatively. The VAS scale is a visual pain scale with values ranging from 0 (no pain) to 10 (maximum pain). This scale comprises four variables corresponding to the areas of pain (lumbar, gluteus, leg, and foot). The ODI is an instrument used to describe pain or limitations. Ten sections are analyzed, each containing six items that describe an increasing degree of severity, with a score of zero indicating little or no pain or functional limitations. In contrast, a score of five indicates extreme pain or restriction. The ODI is composed of 10 variables corresponding to each of the sessions from I to X.

The MacNab scale assessed satisfaction with surgery with values from 1 to 4, with 1 meaning excellent, 2 meaning good, 3 meaning regular, and 4 meaning poor. The surgical procedure was considered unsuccessful when the patient had low back or radicular pain (VAS > 5), associated with a moderate or high physical disability (ODI > 20%) at any time during the 24-month evaluation period. Recurrence of lumbar disk herniation (LDH) was defined by the appearance of symptoms associated with lumbar disk herniation imaging on MRI after an asymptomatic 2-week period after surgery.

### 2.3. Statistical Methods

To check the difference in each scale’s values according to time, the generalized equations estimating (GEE) method [18] was used, which accounts for the correlation between the repeated measures of the same individuals. The GEE method is known as a marginal model. It can be considered an extension of the generalized linear models [19], which directly incorporate the correlation between the measurements of the same sample unit. Initially, the models were performed in a univariate manner. An analysis of variance (ANOVA) was performed between the models of the variables of interest with the model that considered all the periods observed; this provided the *p*-value for each variable’s relationship with a scale over time. Models were then created for each period, accounting for the reference period’s interaction and the relationship with other periods. The mean and standard deviation (SD) were used to describe numerical characterization variables. Absolute and relative frequencies were used for each category. We used R software (version 3.5.0) for statistical analyses, with a significance level of *p* ≤ 0.05. 

## 3. Results

However, the difference between the percentages was minimal. The patients’ mean age was 51.50 ± 14.68 years. The most frequent diagnosis was herniated disk with lateral recess stenosis (45.7%). The most frequent surgical level among patients was between L4 and L5 (46.6%) and on the left side (51.4%), as seen in Table 1.

There was a significant reduction in pain throughout the evaluated times in the lumbar, gluteal, leg, and foot regions (Table 2; *p* < 0.001). There was a significant reduction in pain in the lumbar and gluteal regions compared to T0 (*p* < 0.001), and the perceived pain continued to decrease statistically between all the evaluated times (*p* < 0.05). On the other hand, for the leg and foot regions, patients reported a significant reduction in pain over time (*p* < 0.001), despite the improvements in this symptom slowing over several months.

The ODI showed a significant improvement throughout the evaluation period (*p* < 0.001). Regarding the comparison between times, there was a statistical reduction in the rates of physical disability at T2, T3, T4, T5, and T6 compared to T1 (*p* < 0.05). There was a significant reduction in the ODI at T4, T5, and T6 compared to T2 (*p* < 0.05). All the data are shown in Table 3.

Regarding the MacNab scale, most patients rated the surgery good and excellent at all times (Table 4).

### Surgical Complications

During the 24-month study period, patients who presented with recurrence of low back or radicular pain (VAS > 5) associated with a moderate or high physical disability (ODI > 20%) in the postoperative period, failure of conservative treatment after 4 weeks, and regular or poor welfare according to the MacNab scale were introduced to new interventions. Of the 247 patients who underwent surgery, 18 (7.3%) had complications (Table 5). The most common complication was dysesthesia (27.8%), followed by hernia recurrence (16.7%), insufficient decompression (11.1%), insufficient diskectomy (11.1%), refractory pain (11, 1%), worsening of degeneration (11.1%), diskitis (5.6%), and dural injury (5.6%). The most frequent surgical approach was L5-S1 (55.6%), and the most affected side was the right side (61.1%). 

Four patients with dysesthesia were clinically treated with analgesics, adjuvants (gabapentinoids and antidepressants), corticosteroids (for seven days), and intensive physical rehabilitation. Of these, one was submitted to a root and ganglion block with an improvement in symptoms. The three recurrence cases were treated according to the patients’ symptoms and functional limitations; one patient underwent only conservative clinical treatment and rehabilitation, one underwent foraminal infiltration, and the third patient with a massive hernia underwent a new endoscopic procedure 12 months after the first surgery. Two patients had insufficient decompression; one was treated clinically, and the other underwent a new endoscopic technique. Two patients had an insufficient diskectomy and underwent a new endoscopic procedure at 1.5 and 6 months after the first procedure. Two patients had refractory axial pain that was difficult to control, with no compressive signs on postoperative examinations, and underwent a facet block. Two patients showed significant worsening of disk degeneration at the approached level and underwent arthrodesis 10 and 14 months after the first procedure. One case of diskitis was submitted to a new endoscopy through the same access for material collection and surgical cleaning. One case of dural injury remained asymptomatic for only 24 h under observation and underwent rehabilitation without complications.

Even patients with complications showed a significant reduction in pain at the moments analyzed in relation to T0 (*p* < 0.001). In the lumbar spine region, we observed a significant reduction in VAS at the time of analysis compared to that at T0 (*p* < 0.001). In addition, the VAS at T4 was significantly lower than that at T1 (*p* = 0.034). There were no significant differences between the other analyzed times. For the gluteal region, the reported pain was statistically lower in T2, T3, T4, and T5 than at T0 (*p* < 0.05). Pain analysis at T3 and T5 was lower than that at T1 (*p* < 0.05). For the analysis of leg pain, there was a significant reduction in relation to T0 (*p* = 0.002), despite the stabilization of the VAS values at the other times analyzed. The reported foot pain was statistically lower in T2, T3, T4, and T5 than at T0 (*p* = 0.001). Foot pain analysis at T3 and T4 was lower than that at T2 (*p* < 0.05). Figure 3 shows the pain evolution curve for each region affected.

Patients with complications showed a significant reduction in ODI at the postoperative time compared to T0 (*p* < 0.001).

## 4. Discussion

A low rate of complications was observed in the present study using the full-endoscopic extraforaminal surgical technique with foraminoplasty. Most cases were resolved using conventional clinical treatments. The high rate of patient satisfaction and early rehabilitation were relevant and presented significant data, preserving the integrity of the posterior vertebral components and minimizing segmental instabilities.

A full-endoscopic extraforaminal approach with foraminoplasty provided an alternative, minimally invasive approach to treating the lumbar spine, preserving important stabilizing muscles, maintaining the patient’s ability to move, and eliminating or delaying the need for fusion [20]. Therefore, the rapid recovery of patients, as well as the reduction in pain and rates of physical disability, were the significant outcomes. A series of findings corroborate our results. In one of the first reports, Yeung et al. [21] reported substantial postoperative results in 86.4% of patients who underwent endoscopic surgery. Recently, Li et al. [22] reported an expressive number of good and excellent evaluations in the MacNab scoring system, with a total success rate of up to 92.5% in 148 patients with non-contained LDH.

We demonstrated a significant and persistent improvement in pain site assessment over the assessment period. Corroborating our results, Li et al. [23] indicated a progressive improvement in pain markers in the lumbar spine and legs after a 2-year follow-up of 85 patients. In addition, other studies with similar techniques showed a progressive improvement in pain markers (VAS) and ODI indexes, with low rates of complications [24,25]. 

Our results showed that 46.6% of the surgeries were performed at L4-L5. Corroborating our results, Li et al. [26] reported data similar to ours (of the 134 cases, 78 were between L4-L5) and Hasan et al. reported even higher rates of L4-L5 involvement (63%) [24]. Even after 5 years of follow-up, Li et al. [26] found that patients continued to report good and excellent scores on the MacNab welfare scale. Similarly, Hasan et al. reported a clinically important difference for VAS leg pain and ODI of approximately 90% and 88%, respectively, after 2 years of follow-up [24]. Qiao et al. also reported great results, in which VAS and ODI significantly decreased after 1 year of follow-up [25]. 

According to the literature, transforaminal lumbar endoscopic diskectomy with or without foraminoplasty is a safe and effective treatment for LDH, with a low complication rate [24,25,27]. Damage to the nerve root or dorsal ganglion is the most common complication, with an estimated incidence rate between 1% and 8.9%, especially when dysesthesia is the primary clinical sign [28]. The main nerve damage mechanism is direct injury or compression of the nerve root, caused by the cannula during introduction into the intervertebral foramen. Recently, Hua et al. [29] reported a rate of neurological complications of only 2.1% for surgery at the L4-L5 level using the transforaminal technique with foraminoplasty under general anesthesia. The addition of foraminoplasty to the procedure aims to help expand the foraminal space and improve the safety of the working cannula placement. When transforaminal access is performed under local anesthesia and sedation, neural irritation can be detected early, and the cannula must be repositioned.

To ensure clinical success of this minimally invasive surgical intervention, three important details are crucial: (1) the size of the patient and the point of entry of the instruments into the skin; (2) the location of the herniated disk, its degree of extrusion, and its migration characteristics and fragmentation; and (3) the dimensions of the conjugation foramen [30].

Dura injury is another complication associated with lumbar endoscopy and is usually related to instruments, especially Kerrison. Inadvertent punctures of the dura mater most commonly produce low-output fistulas, and no specific treatment is necessary. When removing the working cannula, the musculature closes around the created path as there are no virtual cavities, and there is no formation of fistulas or hygromas. The fistula closes naturally after pressure equalization inside and outside the dural sac, and no further surgical intervention is necessary. According to the literature, this complication has a low incidence in the transforaminal technique [27,31,32].

The reoperation rate has been reported to range from 2.3% to 15%. Choi et al. [33] evaluated 10,228 endoscopic procedures in the lumbar region and observed a 4.3% failure rate, with incomplete removal of the herniated disk and early recurrence being the most common causes. They identified that the relationship between the working cannula’s positioning and the location of the herniated disk influenced clinical results. Of the 436 patients who underwent reoperation within 6 weeks, 326 (74.8%) underwent open diskectomy, 108 (24.8%) underwent new endoscopic surgery, and 2 underwent arthrodesis (0.5%) [31]. Several studies have shown the recurrence rate of lumbar endoscopic diskectomy (up to 7.4%), similar to microdiskectomy (ranging from 1% to 21%) [34].

Spondylodiscitis after lumbar endoscopic surgery is a relatively uncommon complication of microdiskectomy [25], which requires a smaller incision, less tissue damage, and a continuous flow of saline during the endoscopic procedure. Several hypotheses have been suggested to explain the occurrence of infection after percutaneous lumbar endoscopic diskectomy, such as the higher frequency of placement of the instruments through the cannula combined with a longer surgical time at the beginning of the learning curve; initial puncture of contaminated skin; accidental puncture of the intestine; perforation of the surgical glove after several manipulations with the forceps, cannula, and dilator; and contamination with the fluoroscopy device.

Vascular damage to structures anterior to the intervertebral disk and visceral injury are rare complications. Abdominal vessel injury can occur after inadvertent slipping of instruments, especially when approaching extraforaminal herniated disks when the skin puncture occurs more medially. On the other hand, peritoneal perforation can occur when the point of entry into the skin is in an extreme-lateral position, which is associated with the trajectory of the most verticalized puncture.

The complication rate in this study was 7.3%. These complications were mostly transient and mild to moderate in severity, with 11 of the 18 patients receiving clinical treatment or an anesthetic. Seven patients had a surgical outcome, of which five required another operation using the same technique, and two required arthrodesis. All cases in this study had favorable clinical outcomes.

In this study, we determined the average VAS between the types of complications over time (Figure 3). It was observed that, in diskitis, the pain was already accentuated in the first few weeks. During decompression or insufficient diskectomy, the patients’ pain levels improved in the immediate postoperative period. However, the pain increased less than one month after the procedure. In patients with dysesthesia, clinical improvement occurred one month after surgery on average. When comparing the VAS behavior between locations (lumbar, gluteus, leg, and foot), there was a statistically significant difference between the lumbar region and the foot. Thus, the rate of reduction in the VAS was higher in the foot and lower in the lumbar region, which indicates that pain in the lumbar reduction reduces more slowly.

No vascular injury, injury to visceral organs, and permanent neurological injury were observed. There was one case in which spondylodiscitis reopened via the transforaminal endoscopic technique for cleaning and collecting material for culture, and the patient received subsequent treatment with antibiotics for 4 weeks.

Performing endoscopic diskectomy and foraminoplasty is dependent on the surgeons’ experience, which can be acquired for each technique via a learning curve. This requires at least 30 cases per operated level [27], thorough preoperative planning, and the correct identification and removal of fragments. In these cases, both the incorrect positioning of the needle and the dilator and the working channel being very lateral to the possible area of access to the hernia will provide access to the pulpous nucleus in a portion anterior to the region of hernia extrusion. This makes it difficult to remove the hernia and its fragments, especially for high-grade or migrated hernias [28]. A herniated disc may be contained or non-contained material. A contained disc herniation is said to occur when the displaced portion is covered by the outer annular muscle. If there is no such covering, it is referred to as non-contained. In most cases of non-contained disc herniation, there is a large amount of displaced disc material, which is often sequestered and migrated. It is also difficult to remove successfully by surgical treatment [16]. In addition, an insufficient or neglected foraminoplasty makes it impossible to remove the hernia, which generates inadequate decompression of the vertebral canal, removal of the pulpal nucleus unrelated to the root disk conflict, loss of disk height, and possible worsening of the degeneration and associated segmental instability. These details may also be responsible for residual symptoms, such as dysesthesia, refractory pain, recurrence, and worsening of degeneration.It is important to note, however, that although we have presented very promising results with good clinical significance, our study does not present an equivalent control group and lacks blinding targets (patients) or executors (surgeons). These measures are fundamental to achieving a better comparative effect with greater statistical importance, and their absence may create some limitations in the direct interpretation of our results.

## 5. Conclusions

Based on our results, we conclude that the extra-articular full-endoscopic lumbar technique with foraminoplasty is efficient and less invasive than alternative strategies. The safety of the procedure is very well indicated in cases of centrolateral/lateral herniation or stenosis for patients with symptoms of radiculopathy, and for those who did not respond to conventional rehabilitation treatment or chronic pain management; it resulted in few complications, most of which were treated clinically.

## Figures and Tables

**Figure 1 jpm-13-00453-f001:**
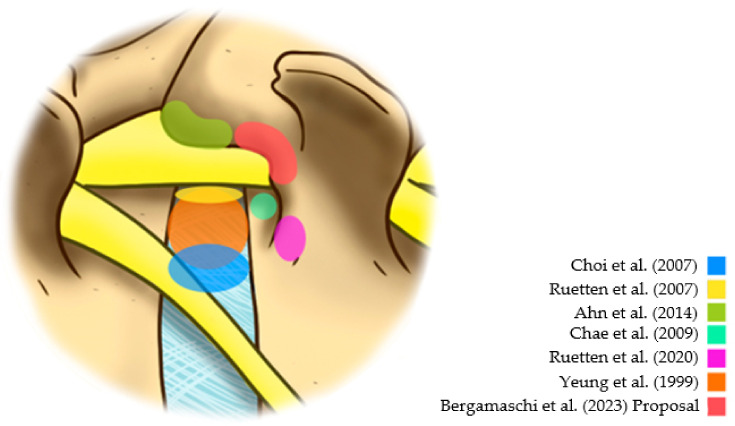
Different reference points in the posterolateral approaches [7,10,11,12,13,16].

**Figure 2 jpm-13-00453-f002:**
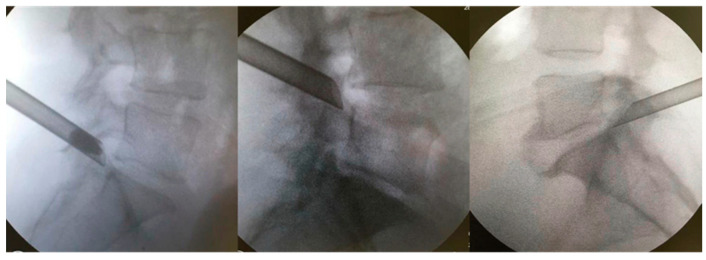
Extraforaminal position of the working cannula before foraminoplasty is supported by the caudal vertebra’s SAP.

**Figure 3 jpm-13-00453-f003:**
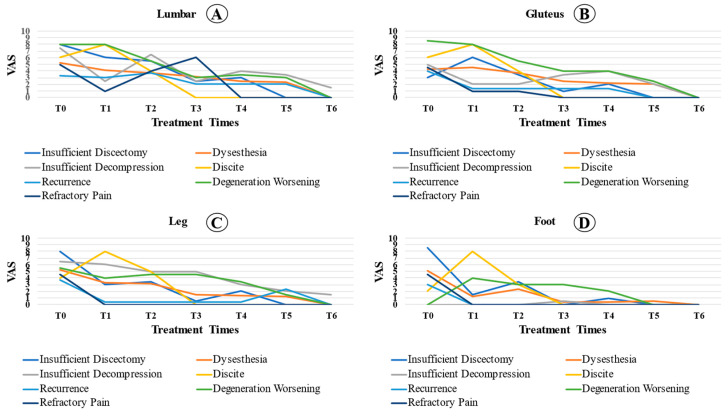
Comparison of the complicated patient’s VAS between locations ((**A**): lumbar; (**B**): gluteus; (**C**): leg; (**D**): foot) over the preoperative time (T0), 2 days (T1), 1 month (T2), 3 months (T3), 6 months (T4), 12 months (T5), and 24 months (T6) postoperative period among cases with complications.

**Table 1 jpm-13-00453-t001:** Descriptive analyses of the sample characterization variables.

Variables		Mean ± SD
**Age (years)**		51.50 ± 14.68
Diagnostic	Hernia + lateral recess stenosis	113 (45.7%)
	Centrolateral disc herniation contained	80 (32.4%)
	Foraminal and lateral recess stenosis	23 (9.3%)
	Centrolateral extruded disc herniation	18 (7.3%)
	Lateral recess stenosis	10 (4.1%)
	Foraminal stenosis	3 (1.2%)
Level	L4-L5	115 (46.6%)
	L5-S1	50 (20.2%)
	L4-L5/L5-S1	35 (14.2%)
	L3-L4/L4-L5	27 (10.9%)
	L3-L4	11 (4.5%)
	L2-L3/L3-L4	3 (1.2%)
	L3-L4/L4-L5/L5-S1	3 (1.2%)
	L2-L3	1 (0.4%)
	L2-L3/L5-S1	1 (0.4%)
	L2-L3/L3-L4/L4-L5	1 (0.4%)
Side	Left	127 (51.4%)
	Right	120 (48.6%)

**Table 2 jpm-13-00453-t002:** Comparison of VAS values by region over the preoperative time (T0), 2 days (T1), 1 month (T2), 3 months (T3), 6 months (T4), 12 months (T5), and 24 months (T6) postoperatively.

Region	Time	Mean ± SD	*p*-Value					
			T0	T1	T2	T3	T4	T5
Lumbar N = 247	**T0**	5.85 ± 2.42	-	-	-	-	-	-
	T1	2.60 ± 2.17	**0.001**	-	-	-	-	-
	T2	2.48 ± 2.11	**0.001**	0.420	-	-	-	-
	T3	1.93 ± 1.92	**0.001**	**0.001**	**0.001**	-	-	-
	T4	1.36 ± 1.56	**0.001**	**0.001**	**0.001**	**0.001**	-	-
	T5	1.31 ± 1.49	**0.001**	**0.001**	**0.001**	**0.001**	0.690	-
	T6	1.22 ± 1.46	**0.001**	**0.001**	**0.001**	**0.001**	0.180	0.652
Gluteus N = 247	T0	4.84 ± 3.16	-	-	-	-	-	-
	T1	1.72 ± 2.31	**0.001**	-	-	-	-	-
	T2	1.51 ± 2.02	**0.001**	0.129	-	-	-	-
	T3	1.21 ± 1.70	**0.001**	**0.001**	**0.006**	-	-	-
	T4	0.90 ± 1.37	**0.001**	**0.001**	**0.001**	**0.001**	-	-
	T5	0.87 ± 1.31	**0.001**	**0.001**	**0.001**	**0.001**	0.891	-
	T6	0.77 ± 1.31	**0.001**	**0.001**	**0.001**	**0.001**	0.082	**0.018**
Leg N = 247	T0	2.54 ± 3.02	-	-	-	-	-	-
	T1	0.68 ± 1.61	**0.001**	-	-	-	-	-
	T2	0.53 ± 1.28	**0.001**	0.100	-	-	-	-
	T3	0.26 ± 0.90	**0.001**	**0.001**	**0.001**	-	-	-
	T4	0.22 ± 0.74	**0.001**	**0.001**	**0.001**	0.339	-	-
	T5	0.21 ± 0.78	**0.001**	**0.001**	**0.001**	0.453	0.879	-
	T6	0.22 ± 0.74	**0.001**	**0.001**	**0.001**	0.593	0.818	0.727
Foot N = 247	T0	5.53 ± 2.73	-	-	-	-	-	-
	T1	1.71 ± 2.18	**0.001**	-	-	-	-	-
	T2	1.44 ± 2.06	**0.001**	**0.029**	-	-	-	-
	T3	0.82 ± 1.53	**0.001**	**0.001**	**0.001**	-	-	-
	T4	0.61 ± 1.31	**0.001**	**0.001**	**0.001**	**0.020**	-	-
	T5	0.53 ± 1.18	**0.001**	**0.001**	**0.001**	**0.001**	0.356	-
	T6	0.52 ± 1.17	**0.001**	**0.001**	**0.001**	**0.001**	0.340	0.839

**Table 3 jpm-13-00453-t003:** Comparison of ODI score over the preoperative times (T0), 2 days (T1), 1 month (T2), 3 months (T3), 6 months (T4), 12 months (T5), and 24 months (T6) postoperatively.

Time	Mean ± SD	*p*-Value					
		T0	T1	T2	T3	T4	T5
**T0**	35.40 ± 15.40	-	-	-	-	-	-
T1	21.00 ± 17.00	**0.001**	-	-	-	-	-
T2	15.80 ± 13.90	**0.001**	**0.001**	-	-	-	-
T3	13.00 ± 12.70	**0.001**	**0.043**	**0.001**	-	-	-
T4	12.60 ± 12.30	**0.001**	**0.001**	**0.001**	0.536	-	-
T5	12.20 ± 11.30	**0.001**	**0.001**	**0.001**	0.288	0.361	-
T6	12.10 ± 11.10	**0.001**	**0.001**	**0.001**	0.220	0.279	0.652

**Table 4 jpm-13-00453-t004:** Comparison of the MacNab scale according to the periods 1 month (T2), 3 months (T3), 6 months (T4), 12 months (T5), and 24 months (T6) postoperatively.

Time	Excellent	Good	Regular	Poor
T2	82 (33.2%)	132 (53.5%)	31 (12.5%)	2 (0.8%)
T3	110 (44.9%)	111 (45.3%)	22 (9%)	2 (0.8%)
T4	110 (45.4%)	109 (45.1%)	21 (8.7%)	2 (0.8)
T5	114 (47.7%)	108 (45.2%)	16 (6.7%)	1 (0.4%)
T6	114 (48.1%)	107 (45.2%)	15 (6.3%)	1 (0.4%)

**Table 5 jpm-13-00453-t005:** Description of complicated cases according to gender (M = male/F = female), age (years), primary diagnosis, surgical level, type of complication, and outcome.

Gender	Age	Primary Diagnosis	Level	Complications	Outcome
M	21	Centrolateral extruded disk herniation	L4-L5/L5-S1	Insufficient Diskectomy	Reoperation (6 months PO)
F	53	Centrolateral disk herniation contained	L5-S1	Dysesthesia	Clinical
F	50	Herniated disk and lateral recess stenosis	L4-L5/L5-S1	Dysesthesia	Clinical
M	39	Centrolateral extruded disk herniation	L5-S1	Dysesthesia	Clinical
F	34	Centrolateral extruded disk herniation	L5-S1	Insufficient Diskectomy	Reoperation (1.5 months PO)
F	74	Foraminal and lateral recess stenosis	L3-L4/L4-L5/L5-S1	Insufficient Decompression	Reoperation (12 months PO)
M	48	Centrolateral disk herniation contained	L4-L5/L5-S1	Diskite	Reoperation (1.5 months PO)
F	62	Foraminal and lateral recess stenosis	L5-S1	Dysesthesia	Foraminal block (PO 12 months)
M	50	Herniated disk and lateral recess stenosis	L4-L5/L5-S1	Hernia recurrence	Reoperation (12 months PO)
M	49	Lateral recess stenosis	L4-L5/L5-S1	Insufficient Decompression	Clinical
M	25	Centrolateral disk herniation contained	L5-S1	Dysesthesia	Clinical
M	68	Centrolateral disk herniation contained	L2-L3/L5-S1	Hernia recurrence	Foraminal block (6 months PO)
F	72	Centrolateral disk herniation contained	L5-S1	Dural Injury	Clinical
F	61	Herniated disk and lateral recess stenosis	L4-L5	Degeneration Worsening	Arthrodesis (10 months PO)
F	67	Herniated disk and lateral recess stenosis	L4-L5	Degeneration Worsening	Artrodesis (14 months PO)
F	32	Centrolateral extruded disk herniation	L4-L5	Hernia recurrence	Clinical
M	32	Centrolateral disk herniation contained	L4-L5	Refractory Pain	Facet Blocking (2 months PO)
M	69	Herniated disk and lateral recess stenosis	L4-L5/L5-S1	Refractory Pain	Facet Blocking (3 months PO)

## Data Availability

The research data will be made available upon request by the corresponding author due to privacy.

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
