# Peer review of "Extraforaminal Full-Endoscopic Approach for the Treatment of Lateral Compressive Diseases of the Lumbar Spine"

_jpm, 2023, doi:10.3390/jpm13030453_

Round 1
Reviewer 1 Report
It is well designed manuscript. It is suitable to publication
Author Response
São Paulo, February 10th, 2023.
Dear Reviewer,
We would like to thank the referees´ suggestions for the article entitled "Extraforaminal Full-Endoscopic Approach for the Treatment of Lateral Compressive Diseases of the Lumbar Spine", which undoubtedly contributed to improving our article prior to publication in the “Journal of Personalized Medicine”. We apologize for all mistakes, mainly English language mistakes. An English review was carried out by a native English speaker (see the certificate at the end of the document). We hope that we have addressed all the reviewer’ concerns and that our revised chapter can be accepted in its present version. All alterations performed in the manuscript are highlighted in yellow and the changes are tracked. Thank you very much for your review and opinion.
Sincerely,
João Paulo M. Bergamaschi

Reviewer 2 Report
In this paper the authors reported Full-endoscopic transforaminar approach for extraforaminar disc herniation. We know first inside-out technique was recommended but now outside-in with foraminotomy is major method. They reviewed the history of the FESS and I agree with them. But I cannot understand the difference between TELD with foraminotomy and their extraforaminal extra-articular approach. What did they mean “extra-articular”? Many authors reported many technical terms. Finally, AOSpine showed the consensus paper (Global spine journal 2020; 10(25); 111S-21S). In their classification, this method should be called transforaminal endoscopic lumbar foraminotomy and extraforaminal endoscopic lumbar discectomy depends on the pathologies.
And anatomical words should be checked.
And this article will be published special issue. I found publication in English about Extraforaminal extraarticular endoscopic access with foraminoplasty in the spine journal. In that journal cases in 2016-2018 were reported. This time the authors described the cases in 2018-2020, but case number is same 247 and complication rate is same. I cannot decide these means duplication or not. I’ll leave it to the editorial office.

Author Response
São Paulo, February 10th, 2023.
Dear Reviewer,
Thank you for the comments. We analyzed carefully point-by-point.
- The term “extra-articular” was removed from the title and from the whole text, because it was, in fact, redundant. The correct term for the approach is extraforaminal full-endoscopic approach.
- The anatomical words were revised by a native English-speaking colleague. You can find the certificate attached at the end of the manuscript.
- Finally, the article published in The Spine Journal is only an abstract submitted to NASS 2022 Annual Meeting, not the full paper. In fact, the cases occurred from 2016 to 2018 and not from 2018 to 2020. This was corrected in the Methods. The full paper (with tables, graphics and more details of the study) is presented here.
Sincerely,
João Paulo M. Bergamaschi

Reviewer 3 Report
I congratulate the authors for reporting their results in this uniportal endoscopic trans-articular approach.
The methodology is adequate, and despite being a retrospective study, the inclusion criteria are specific, and the conclusions are proper. I have very few comments about it that will surely enrich your manuscript:
1.- What is the difference between the approach proposed by the authors and the one reported by Hofstetter et al.
Hasan S, White-Dzuro B, Barber JK, Wagner R, Hofstetter CP. The Endoscopic Trans-Superior Articular Process Approach: A Novel Minimally Invasive Surgical Corridor to the Lateral Recess. Opera Neurosurg (Hagerstown). 2020;19(1):E1-E10. doi:10.1093/ons/opaa054
2.- It would be appropriate to cite the articles that have recently reported results with this approach:
Hasan S, White-Dzuro B, Barber JK, Wagner R, Hofstetter CP. The Endoscopic Trans-Superior Articular Process Approach: A Novel Minimally Invasive Surgical Corridor to the Lateral Recess. Opera Neurosurg (Hagerstown). 2020;19(1):E1-E10. doi:10.1093/ons/opaa054
Qiao L, Liu JY, Tang XB, et al. The Trans-Superior Articular Process Approach Utilizing Visual Trephine: A More Time-Saving and Effective Percutaneous Endoscopic Transforaminal Lumbar Discectomy for Migrated Lumbar Disc Herniation. Turk Neurosurg. 2022;32(4):612-617. doi:10.5137/1019-5149.JTN.34049-21.3
The rest of the manuscript is suitable for publication in the journal.
Author Response
Dear Reviewer,
Thank you very much for your comments. We checked point-by-point and our reply is below. The paper was checked and some modifications were done. An English editing was done by a native speaker colleague. The certificate is attached at the end of the manuscript.
1 – The difference is that we do not use fluoroscopic percutaneous trephines to introduce the work sleave. Initially, we introduced the work sleave and, under direct vision, we performed the foraminoplasty with a drill. We believe that in this way the procedure becomes safer and can be used in cases of very small Kambin's triangle. This part was described on lines 111 to 114.
2 – The articles suggested were cited in the text (from line 265 to line 277). Here is the currently text and citations:
“We demonstrated a significant and persistent improvement in pain site assessment over the assessment period. Corroborating our results, Li et al. [22] indicated a progressive improvement in pain markers in the lumbar spine and legs after a 2-year follow-up of 85 patients. In addition, other studies with similar techniques showed a progressive improvement in pain markers (VAS) and ODI indexes, with low rates of complications [23, 24].
Our results showed that 46.6% of the surgeries were performed at L4-L5. Corroborating our results, Li et al. [25] reported data similar to ours (of the 134 cases, 78 were between L4-L5) and Hasan et al. reported even higher rates of L4-L5 involvement (63%) [23]. Even after 5 years of follow-up, Li et al. [25] found that patients continued to report good and excellent scores on the MacNab welfare scale. Similarly, Hasan et al. reported a clinically important difference for VAS leg pain and ODI of approximately 90% and 88%, respectively, after 2 years of follow-up [23]. Qiao et al. also reported great results, in which VAS and ODI significantly decreased after 1 year of follow-up [24]."
Sincerely,
João Paulo M. Bergamaschi
Round 2
Reviewer 2 Report
The authors changed well and I know this study was completely same data as the previous abstract. This is the matter of editorial office. Thank you.